# Research on the Properties of Steel Slag with Different Preparation Processes

**DOI:** 10.3390/ma17071555

**Published:** 2024-03-28

**Authors:** Xingbei Liu, Chao Zhang, Huanan Yu, Guoping Qian, Xiaoguang Zheng, Hongyu Zhou, Lizhang Huang, Feng Zhang, Yixiong Zhong

**Affiliations:** 1School of Traffic and Transportation Engineering, National Engineering Research Center of Highway Maintenance Technology, Changsha University of Science & Technology, Changsha 410114, China; l1191815148@126.com (X.L.); zc11@stu.csust.edu.cn (C.Z.); guopingqian@csust.edu.cn (G.Q.); zhouhongyu23@126.com (H.Z.); huanglz@stu.csust.edu.cn (L.H.); zhangfeng122333@163.com (F.Z.); 2Shanghai Municipal Engineering Design & Research Institute (Group) Co., Ltd., Shanghai 200092, China; zhengxiaoguang@smedi.com

**Keywords:** preparation process, steel slag, expansion test, XRD, microscopic morphology, thermal stability

## Abstract

To promote the resource utilization of steel slag and improve the production process of steel slag in steelmaking plants, this research studied the characteristics of three different processed steel slags from four steelmaking plants. The physical and mechanical characteristics and volume stability of steel slags were analyzed through density, water absorption, and expansion tests. The main mineral phases, morphological characteristics, and thermal stability of the original steel slag and the steel slag after the expansion test are analyzed with X-ray diffractometer (XRD), scanning electron microscope (SEM), and thermogravimetric analysis (TG) tests. The results show that the composition of steel slag produced by different processes is similar. The main active substances of other processed steel slags are dicalcium silicate (C_2_S), tricalcium silicate (C_3_S), CaO, and MgO. After the expansion test, the main chemical products of steel slag are CaCO_3_, MgCO_3_, and calcium silicate hydrate (C-S-H). Noticeable mineral crystals appeared on the surface of the steel slag after the expansion test, presenting tetrahedral or cigar-like protrusions. The drum slag had the highest density and water stability. The drum slag had the lowest porosity and the densest microstructure surface, compared with steel slags that other methods produce. The thermal stability of steel slag treated by the hot splashing method was relatively higher than that of steel slag treated by the other two methods.

## 1. Introduction

Steel slag is a lumpy substance that combines the residual solvent and metal oxides in the steelmaking furnace after cooling, iron removal, crushing, etc. It is a by-product of the steelmaking process. Steel slag emissions make up about 8–15% of crude steel [1,2]. China’s crude steel production exceeded 1 billion tons and the amount of slag discharge exceeded 120 million tons in 2021, with the strong growth of China’s iron and steel industry [3]. However, the physical mechanics, morphology, and volumetric stability of steel slag are limited by the production equipment, preparation process, and geographical environment, resulting in the overall utilization rate of steel slag being less than 30% [4]. Most steel slag is treated as industrial waste and piled up in the mountains, resulting in land occupation, environmental pollution, resource waste, and other issues [5,6]. In the context of “carbon peak, carbon neutral”, promoting the efficient utilization of steel slag resources is an urgent need to ensure the stable development of China’s society [7].

The research on the material properties of steel slag is the primary problem of its utilization. Experts have made some progress focusing on its physical and chemical properties, mechanical properties, morphological characteristics, and other properties. In terms of the physical and chemical properties of steel slag, experts believe that, through the analysis of the vast majority of steel slag, the main components can be identified as calcium, magnesium, silicon, and iron oxides, so the steel slag can be regarded as a CaO–SiO_2_–MgO–FeO tetrameric system [8,9]. Calcium, silicon, and phosphorus oxide content were used to indicate the alkalinity of steel slag [10]. According to the differences in alkalinity, steel slag can be divided into three categories: high-alkalinity steel slag (M > 2.5), medium-alkalinity steel slag (1.8 < M < 2.5), and low-alkalinity steel slag (M < 1.8). It has been pointed out that the alkalinity of steel slag can reflect its hydration activity [11]. However, it is more reliable to evaluate the hydration activity of steel slag using the activity index rather than alkalinity [12,13]. In addition, Li et al. [14] attempted to evaluate the hydration activity of steel slag materials using chemical methods, and the experimental process operation was more difficult to control. The experts affirmed the physical and mechanical properties of steel slag materials, and the hardness, adhesion, and roughness of steel slag became advantages for certain applications. In addition, some researchers used two-dimensional and three-dimensional image analysis techniques to characterize the morphology of steel slag aggregates [15]. Scanning electron microscopes can also be used to observe the micro-morphological characteristics of steel slag; steel slag has a rough surface texture, and steel slag contains a lot of pores, making it a somewhat porous material [16].

The material properties of steel slag determine its resource uses and application effects. Steel slag contains CaO, Fe, MnO, MgO, Fe_2_O_3_, etc. It can be used as raw material for iron and steel sintering, and its magnesium and calcium composition means it can be used as a solid solution, which can replace part of magnesite, limestone, and other fluxes [17,18,19]. Steel slag contains about 10% of the metal iron, which, after crushing, screening, magnetic separation, and other processing, can be sorted into different particle sizes, of slag steel and magnetically separated powder, for reuse [20]. Steel slag’s main composition is CaO, Al_2_O_3_, SiO_2_, and MgO, which is similar to the raw materials for the traditional construction of tiles; therefore, steel slag can be used as a raw material in the preparation of ceramics [21]. Steel slag contains many alkaline oxides, such as CaO and other compositions, which can be used to treat pollutants in wastewater through chemical reactions [22,23]. Steel slag contains high levels of Si, Mn, and P and various trace elements, which can provide nutrients needed for crop growth. At the same time, CaO in steel slag can also slowly neutralize soil, so it can be used as farmland fertilizer and for improvement purposes [24,25]. Steel slag contains dicalcium silicate and tricalcium silicate, and, as silicate cement clinker has a similar mineral phase, steel slag can be used as a raw material for the production of cement, but also can be applied to concrete admixture, dry mortar, and so on. However, the low content of practical cementitious components in steel slag and its high formation temperature of up to 1600 °C, which results in more crystallization of the mineral phase, limit the broad application of steel slag in cement and concrete [26,27]. Steel slag, crushed stone, and its natural aggregates have similar physico-mechanical properties [28,29]. The density of steel slag is high, generally above 3.2 g/cm^3^, while the density of natural aggregates is about 2.6–2.9 g/cm^3^; steel slag has the advantage of physico-mechanical indices, which show that its crushing value and Los Angeles abrasion are higher than that of natural aggregates by 4.7–20.5% and 12.0–52.7%, respectively. The porous nature of steel slag results in high water absorption levels of steel slag aggregate, generally between 1% and 2% [30,31]. Steel slag crushed stone is suitable for road material requirements, as it can be used for the road base layer, the surface layer, and the bedding, and can also be used as engineering backfill material [32,33,34,35]. The common uses of steel slag are shown in Table 1. Many studies have shown that using steel slag as a road construction material is the most effective means of resource utilization [36,37,38]. However, the volumetric stability of steel slag materials, preparation process limitations, distribution of iron and steel enterprises, policies, and other reasons still limit the use of steel slag in road construction [28,39,40].

Meanwhile, the treatment processes of different steel slags result in significant differences in their performance. Analyzing and comparing the steel slag produced by various methods will be beneficial for promoting the resource utilization of steel slag. Therefore, this study investigates the properties of steel slag (XG/PG/BG/WG) with different preparation processes. In this research, XG and PG are produced using the thermal smothering method, WG is produced using the hot splashing method, and BG is made using the drum method. Hot smothering and drum method steel slag are in a high-temperature steam environment. At the same time, the hot splashing method of steel slag is in a natural environment. Different processing techniques lead to differences in the performance of steel slags. Thus, this study further analyzes steel slags’ physicomechanical characteristics and volumetric stability by density, water absorption, crushing value, and swelling tests. The main mineral phases, morphological characteristics, and thermal stability of steel slags are analyzed using XRD, SEM, and TG tests before and after expansion tests. The results can provide a reliable reference for the application selection of steel slag.

## 2. Materials and Methods

### 2.1. Materials

Steel slags from four steel mills are used in this research: XG, PG, BG, and WG. XG has been stored for more than 12 months, and PG, BG, and WG have been stored for about 3–6 months. After sieving, particle sizes of steel slag range from 4.75 to 16 mm and are classified into 4.75–9.5 mm, 9.5–13.2 mm, and 13.2–16 mm. We take classified steel slags as the test material. The chemical element composition of the four types of steel slag is shown in Table 2. The picture of four kinds of steel slag is shown in Figure 1.

XG and PG steel slags are produced using the hot smothering method. The production process of hot smothering slags is to introduce high-temperature liquid steel slags into a special hot slag pond, cool them with water spray, cover the slag pond to maintain pressure, and use the thermal stress generated by the rapid cooling of steel slags and saturated high-temperature steam to penetrate steel slags. Thus, free calcium oxide in steel slags can be digested to generate calcium hydroxide, resulting in volume expansion and tiny steel slag particles conducive to recovery [45].

BG steel slag is produced using the drum method. The drum slag process involves the granulation of molten slag into the drum at an appropriate flow rate, under rapid cooling conditions, under centrifugal force, spray water, and the squeezing friction of steel balls inside the drum [46]. This process has the advantages of small pollution, clean site, high treatment efficiency, uniform particle size of treated steel slag, stable free calcium oxide, and direct utilization of steel slag. However, the requirements for the iron content of the treated steel slag are strict, and the slag with molten steel is prone to explosion. The equipment is severely worn, and the cost of slag treatment is high [47]. Additionally, there are requirements for the block size of the treated steel slag. Although drum slag equipment capable of handling solid slag has been developed, limitations still exist, and converter steel slag cannot be fully processed.

WG steel slag is produced using the hot splashing method. The production process of hot splashing slag is to load liquid high-temperature steel slag into a tank, lift the tank with a crane, and evenly pour the liquid steel slag into a shallow plate. After filling the shallow plate with water, spray water for 8–45 h and filter for another 2 h. The average temperature of the steel slag can be cooled to around 50 degrees and then sent to crush and classify [48,49].

### 2.2. Experiments

#### 2.2.1. Density Test

In order to compare the basic physical indexes of different steel slag, the density and water absorption of steel slag are tested. The test process was carried out according to the JTGE42-2005 Code for Testing Aggregates in Highway Engineering [50].

#### 2.2.2. Volume Expansion Test

The expansion test was carried out using GB/T 24765-2009 steel slag for wearing asphalt pavement [51] to compare the volume stability of different steel slags. The four kinds of steel slag are 4.3 kg each, and the proportions of particle size 4.75–9.5 mm, 9.5–13.2 mm, and 13.2–16 mm were 48.63%, 44.52%, and 6.85%, respectively. After the water bath temperature reaches 90 ± 1 °C, the constant temperature is maintained for 6 h, and then the heating and cooling are naturally stopped for 10 cycles. The data were recorded and the expansion rate was calculated.

#### 2.2.3. XRD, SEM, and TG Test

Grind XG, PG, BG, and WG steel slags into powder with a ball mill. The powder of four kinds of steel slag is used to conduct the XRD and TG tests. The crystal structure of the sample was characterized by Rigak SmartLab 9KW X-ray Diffraction (XRD) (Rigaku, Auburn Hills, MI, USA). The wavelength of the X-ray is 1.54 Å (CuKa). The test voltage and current are 45 kV and 200 mA, respectively. The scanning range is 5–75°, the scanning frequency is 10°/min, and the step size is 0.02°. Thermogravimetric analysis (TG) of steel slag material is carried out by the Netzsch STA449F3 synchronous thermal analyzer produced in Selb, Germany. The test atmosphere was a nitrogen environment. The test temperature is from room temperature to 1200 °C, and the heating rate is 5 °C/min. Scanning electron microscopy is used to characterize the surface microstructure of the original steel slag produced by different processes and the steel slag after the expansion test. The electrical signals generated by the photoactive focused electron beam on the sample’s surface were converted into images for on-site measurement. The experimental equipment for the XRD, SEM, and TG tests is shown in Figure 2.

## 3. Results and Discussion

### 3.1. Density and Water Absorption

The density and water absorption are shown in Table 3.

It can be seen from Table 3 that BG has the highest density compared to the other three kinds of steel slags. It can be found that the density of particle size of 4.75–9.5 mm decreases in the order of BG, WG, XG, and PG. The density of particle size of 9.5–13.2 mm decreases in the order of BG, WG, PG, and XG. The density of particle size of 13.2–16 mm decreases in the order of BG, WG, XG, and PG. The apparent density of XG, PG, BG, and WG varies from 3.192 to 3.456. The apparent density of four kinds of steel slag is 19.0–28.9% higher than natural aggregate; the average apparent density of four kinds of steel slag is relatively close to the steel slag from Baotou, with a variation of less than 10% [52].

According to the density of steel slag, it can be found that the steel slag produced by the drum method from BG has the highest apparent density. Relative to drum method slag, the rougher production process of hot smothering steel slag and hot splashing slag reduces the density of steel slag.

### 3.2. Volume Expansion

The expansion test results of four kinds of steel slag can be seen in Figure 3.

It can be seen from Figure 3 that the volume expansion of steel slag is continuously increasing. Based on the expansion rate of the volume of steel slag specimens on the tenth day, the expansion rate of WG is the highest, at 2.7%, followed by XG, at 1.52%. The expansion rates of BG and PG are relatively close, and both are around 1%. The volume expansion of steel slag varies from 0.5 to 3.5% in the present research; it can be found that the expansion rates of XG, PG, and BG are at a low level, while the volume expansion rate of WG is relatively high [53,54].

### 3.3. Mineral Changes

XRD analysis is conducted on the steel slags before and after the expansion test to study the mineral phase, as shown in Figure 4. In this case, the suffix 0 is the initial steel slag, and the suffix 10 is the steel slag after the expansion test.

Due to the more substantial alkalinity of CaO compared to MgO, a small amount of SiO_2_ will preferentially form dicalcium silicate or tricalcium silicate with CaO at high temperatures [55]. It can be observed from the XRD diffraction patterns of four different steel slags in Figure 3 that the original steel slag contains elements presenting prominent peaks such as C_2_S, C_3_S, CaO, and MgO. These substances are the main active components in steel slag that participate in the aging reaction. It can be observed that the mineral elements in the four types of steel slag have undergone significant changes. CaCO_3_ and C-S-H in XRD peak patterns show an obvious increase after expansion tests. Meanwhile, C_2_S, C_3_S, CaO, MgO, and other alkaline elements show an apparent decrease.

In addition to these similar mineral changes, there are also changes in some elements that are worth exploring. Considering the minimum density of WG in the previous density experiment, although the peak heights of C-S-H and CaCO_3_ in WG vary less, the lower density results in more significant volume expansion of WG in the expansion rate experiment. Considering the maximum density of BG in the previous density experiment, although the peak heights of C-S-H and CaCO_3_ in BG are very high, the higher density results in less volume expansion of BG in the expansion rate experiment. It is worth noting that the peak around the FeO position of BG steel slag decreased after the expansion test. Because the peak around 42° contains MgO and FeO, MgO reacts with CO_2_ to generate MgCO_3_, and the iron element oxidizes to form iron oxide with higher FeO peaks. Although the peaks of MgO and FeO are relatively close, FeO cannot decrease after the expansion test. The MgO content of BG steel slag is much higher than the other three types of steel slag.

The above analysis shows that the main elements of steel slag produced by different steel mills have similar mineral phases, with the main mineral phases being C_2_S, C_3_S, ferrite, and a small amount of CaO and MgO. The iron oxide content in the steel slag produced by XG and PG using the hot smothering process is significantly higher than that produced by WG and BG. Considering the differences in iron recovery processes, this may indicate that there is still room for modification in the iron recovery processes of XG and PG steel plants. WG and BG contain fewer ferrite compounds, suggesting that steel slag’s production process and supporting recovery measures are better. It can be concluded after XRD analysis that the initial steel slag contains a certain content of unstable elements composed of MgO and CaO in the RO phase. After hydration, the main products of the steel slag are still CaCO_3_ and C-S-H.

### 3.4. Simple Quantitative Analysis

A simple quantitative analysis of C2S, C3S, CaO, and MgO mineral phases is performed using Jade software (version 6.0) on the XRD spectra of four initial steel slag types [56]. Simple quantitative analysis is a built-in quantitative analysis function in Jade software. This function calculates the percentage of selected minerals based on parameters such as peak, angle, and area. It is worth noting that the sum of selected mineral proportions in simple quantitative analysis is 100%. The analysis results mainly consider the four elements C_2_S, C_3_S, CaO, and MgO, which account for a relatively large proportion of steel slag, so these four compounds are mainly considered. The sum of the four elements in the figure is 100%. The analysis results are shown in Figure 5.

A simple quantitative analysis is conducted on the main products of the four types of steel slag after the expansion test, and the analysis results are shown in the figure. It can be seen from the graph that the content of C_2_S and C_3_S is the highest, accounting for over 60% of these four minerals, with the content of C_2_S and C_3_S in XG and WG exceeding 80%. Next is PG, which contains 78.8% of C_2_S and C_3_S. The lowest content of C_2_S and C_3_S is BG, accounting for 68.6%. The analysis of the four main alkaline active minerals in steel slag also reveals interesting results, with C_3_S in XG and WG significantly higher than C_2_S by about two times, while in contrast, C_2_S in BG and PG is significantly higher than C_3_S by about two times. Considering the differences between different steel slags and the influence of various factors, such as other raw materials, the impact of the steel slag production process on the differences in C_3_S and C_2_S content deserves further research.

The main components of the RO phase in steel slag are CaO and MgO. From the simple quantitative analysis chart, it can be found that the content of MgO is greater than that of CaO, with the highest MgO content in BG, followed by PG, XG, and WG. The content of CaO decreases in the order of BG, PG, XG, and WG. The content of MgO decreases in the order of BG, PG, XG, and WG.

Figure 6 shows that the most common mineral composition of the main reaction products of PG, BG, and WG steel slag is hydrated calcium silicate after the expansion test. It can be found that among the four types of steel slag, only XG has the highest proportion of CaCO_3_, exceeding the proportion of hydrated calcium silicate. In the three types of steel slag, PG, BG, and WG, the content of hydrated calcium silicate is the highest, accounting for over 50%. The proportion of hydrated calcium silicate after the BG expansion test is the highest, at 61.4%. The high proportion of hydrated calcium silicate indicates that BG has the highest degree of volcanic ash reaction, with more dicalcium silicate and tricalcium silicate participating in the volcanic ash reaction, because the alkalinity of CaO is higher than that of MgO, and the hydroxyl activity in Ca(OH)_2_ of the hydration product is higher than that in Mg(OH)_2_. Figure 5 shows that the proportion of CaO in BG is the highest, indicating this phenomenon.

The above analysis shows that C_2_S and C_3_S are important alkali-active compounds in steel slag, accounting for 70% to 80% of the total. However, further research on the proportion of C_2_S and C_3_S shows no obvious connection with the steel plant process. The specific differences in production processes, temperatures, and raw materials among different steel plants make it difficult to control variables and specifically study the generation conditions of C_2_S and C_3_S. The generation of C-S-H is not related to the content of water-active products such as C_2_S and C_3_S but rather to the alkaline substance CaO in the environment. XG’s higher alkaline environment is conducive to the generation of C-S-H.

### 3.5. Microstructure Changes

Scanning Electron Microscope (SEM) analysis is performed on the surfaces of four types of untreated steel slag and steel slag after expansion testing, and the results are shown in Figure 7. The suffix 0 represents the initial steel slag, while the suffix 10 represents the steel slag after expansion tests.

Figure 7 shows that the initial microstructure of the steel slag surface is similar, with a rough surface and no noticeable mineral crystals visible. However, after the expansion test, obvious mineral crystals appear on the surface of the steel slag, with most of the mineral crystal structures showing tetrahedral or cigar-like protrusions. Based on comprehensive XRD analysis, the main hydration products of steel slag after the expansion test are CaCO_3_ and C-S-H compounds. It can be inferred that this part of the crystal is mainly C-S-H crystal, and CaCO_3_ is mostly a needle-like structure rather than a polyhedral or protruding crystal structure [57].

### 3.6. Thermogravimetric Analysis

The TG results of the four initial steel slags are shown in Figure 8. Figure 8a is the TG data curve, and Figure 8b is the gradient of TG data (DTG). Figure 8a shows that the mass of PG-0, BG-0, XG-0, and WG-0 continues to decrease with the increased temperature. It can be observed that the rapidly decreasing mass in Figure 8a corresponds to the two weight loss peaks around 100 °C and 700 °C in Figure 8b, where a large amount of mass loss occurs near these temperatures. Figure 8b also shows a third weight loss peak around 400 °C. Only the WG-0 sample shows a significant weight loss peak in Figure 8b, as the sample continuously absorbs water to form Ca(OH)_2_ during storage. At around 400 °C, Ca(OH)_2_ will continuously dehydrate to form CaO. The weight loss peak around 400 °C is not apparent due to the small amount of Ca(OH)_2,_ which slows the reaction between CaO and water. WG exhibits a more significant weight loss peak at 400 °C compared to other samples due to the highest CaO content. The most substantial weight loss peak is around 700 °C due to the decomposition of CaCO_3_ at high temperatures to generate CaO and CO_2,_ which causes weight loss [58].

The minimum mass loss rate of WG steel slag indicates higher CaO and lower CaCO_3_ content. The higher CaO and lower CaCO_3_ content suggests that the mineral phase is stable at high temperatures, followed by BG and PG steel slag in the middle position. XG steel slag has the highest mass loss rate because the CaCO_3_ in XG is the highest. The mass loss and mass loss rate are the highest in the 600–750 °C range. Through comprehensive comparison, it can be found that the mass loss rates of four initial steel slags were ranked from small to large: WG > BG > PG > XG.

The TG and DTG results of the steel slags after the expansion test are shown in Figure 9. It can be found from Figure 9a that the mass loss increases from small to large and is classified as WG, BG, PG, and XG. Figure 9a shows that the mass of the four types of steel slag after the expansion test continues to decrease as the temperature increases.

In Figure 9a, it can be seen that the maximum mass loss peak occurs around 700 °C. It can also be observed from Figure 9b that there is a maximum decrease rate in the range of 600–750 °C. The mass loss in this part is composed of CaCO_3_. The second peak of weight loss is between 100 and 200 °C, and the mass loss in this part is mainly free water. Figure 9b shows that a small weight loss peak appears at 400 °C, which is smaller than at 100 °C. The mass loss around 400 °C decomposes Ca(OH)_2_ into solid CaO and gas state H_2_O. Only WG exhibits a relatively small weight loss peak at 400 °C, indicating that the steel slag of WG generates more Ca(OH)_2_ during hydrolysis and reaction. More Ca(OH)_2_ demonstrates that WG steel slag contains more CaO, consistent with the highest WG expansion rate in the expansion test results.

The mineral content in the steel slag has changed after expansion testing. The mass loss and mass loss rate are relatively higher than the results compared to the TG results of the initial steel slag. During the expansion experiment, more CaCO_3_ and C-S-H were generated, which released CO_2_ and bound water during thermal decomposition. Based on the TG test of the initial steel slag and steel slag after the expansion test, it can be concluded that WG steel slag has the highest thermal stability performance, followed by PG, BG, and XG.

Considering that TG and DTG curves can only qualitatively analyze the quality loss of four types of steel slag, they cannot quantitatively analyze the quality changes of steel slag at different temperatures. Therefore, a quantitative analysis was conducted on the mass loss in the three mass loss peak temperature ranges of steel slag before and after the expansion experiment. The temperature range of the weightlessness peak and the mass loss rate was calculated in three intervals: 100–200 °C, 400–500 °C, and 600–750 °C. The statistical data results are shown in Table 4.

The maximum mass loss of XG steel slag occurs between 100 and 200 °C due to the larger pore structure of XG, which has higher voids and water absorption. The mass loss rate of XG steel slag is the highest in the temperature range of 400–500 °C due to the high CaO content in XG steel slag, which generates the most Ca(OH)_2_ in the hydration reaction. The 600–750 °C temperature range mainly involves the decomposition of CaCO_3_ into CaO and CO_2_. The maximum mass loss rate of XG steel slag in this temperature range indicates the highest content of CaCO_3_, followed by PG and BG steel slag, respectively. In the original sample, XG steel slag has the highest CaCO_3_ content and water absorption rate; WG has the lowest water absorption and CaCO_3_ content; PG and BG are in the middle.

It can be seen from Table 5 that XG has the highest mass loss rate in the range of 100–200 °C, indicating that XG steel slag has the highest water absorption rate, followed by PG, BG, and WG. The 400–500 °C temperature range has the highest Ca(OH)_2_ content in XG, followed by PG, WG, and BG. XG has the highest weight loss rate in the 600–750 °C range, followed by PG, BG, and WG. Compared with other research [59,60], it can be found that the thermal stability of four kinds of steel slag is enough for construction. From TG, DTG, and qualitatively analyzed results, it can be concluded that WG has the highest thermal stability, followed by BG, PG, and XG.

## 4. Conclusions

This study compared the physical and chemical properties of steel slag produced by three different methods from four different steel mills. We compared the changes in mineral phases, microstructure, water stability, and thermal stability of steel slag before and after the expansion test. The research results are of great significance for the resource utilization of steel slag and the improvement of steel slag treatment methods in steelmaking plants. The conclusions of this research can be drawn as follows:(1)The steel slag produced by the drum method has the highest density and lower expansion rate than the other three steel slags. It was also found that although BG has the highest content of free calcium oxide, the higher density of steel slag particles helps to resist the volume expansion caused by water contact.(2)The composition elements of steel slag produced by different processes are very similar, with the primary active materials being C_2_S, C_3_S, CaO, and MgO. After the expansion test, the main chemical products of steel slag are CaCO_3,_ MgCO_3_, and C-S-H. C-S-H is generated by hydrolysis and a combination of C_2_S and C_3_S, and the content of CaO and MgO mainly influences the amount generated. This means a higher degree of alkalinity can promote the generation of C-S-H. Most C-S-H is generated on the surface of steel slag, forming clusters of mineral crystals.(3)The thermal stability of the two types of steel slag treated by the hot smothering method is relatively lower than that of steel slag treated by the hot splashing and drum methods. Additionally, it was found that the storage time may be related to the thermal stability of steel slag particles.(4)This research considers the density, volume stability, and thermal stability of the four types of steel slag. XG, PG, and BG are recommended as alternative coarse aggregates for road and building construction. Meanwhile, WG is unsuitable as a building material due to its high expansion, and further treatment should be carried out to reduce its expansion before testing. Among them, BG steel slag has the best comprehensive performance.

There are still some shortcomings in existing research, and in future research, more steel slag produced by existing processes and stricter variable control will be adopted.

## Figures and Tables

**Figure 1 materials-17-01555-f001:**
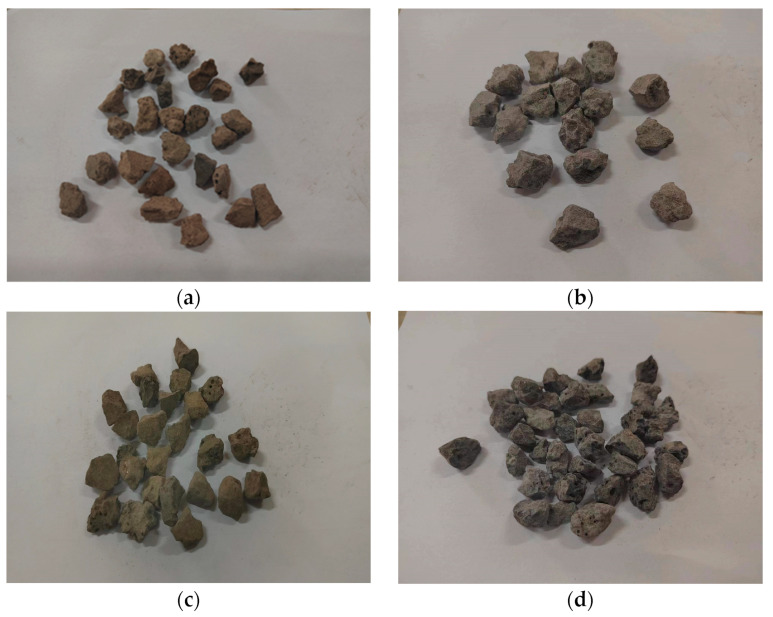
Pictures of steel slags. (**a**) PG; (**b**) BG; (**c**) WG; (**d**) XG.

**Figure 2 materials-17-01555-f002:**
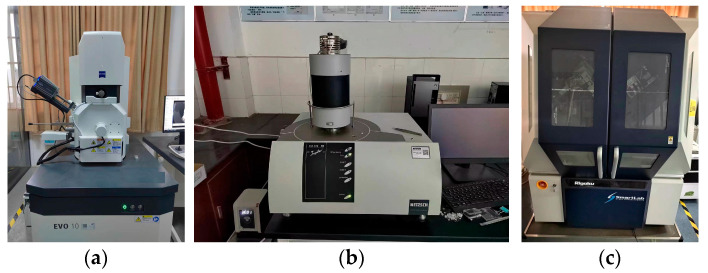
Experimental equipment: (**a**) SEM test device; (**b**) TG test device; (**c**) XRD test device.

**Figure 3 materials-17-01555-f003:**
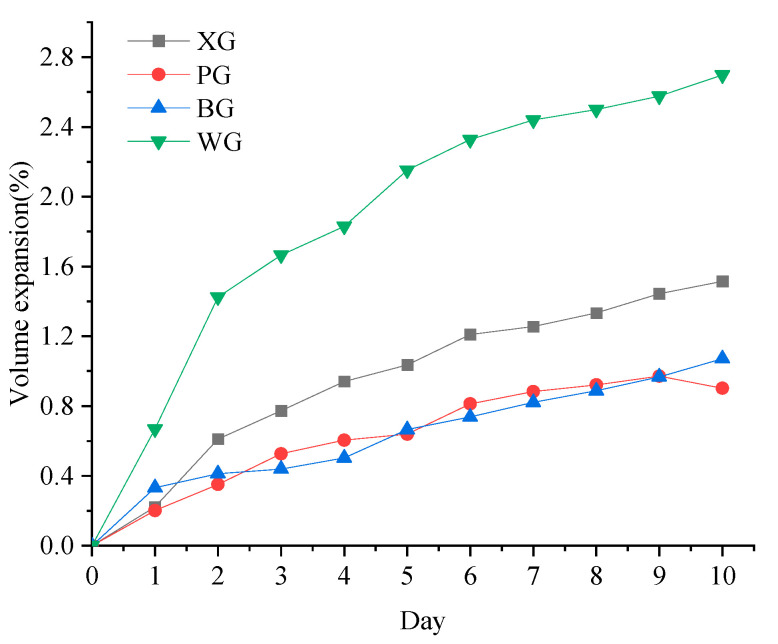
The expansion test results.

**Figure 4 materials-17-01555-f004:**
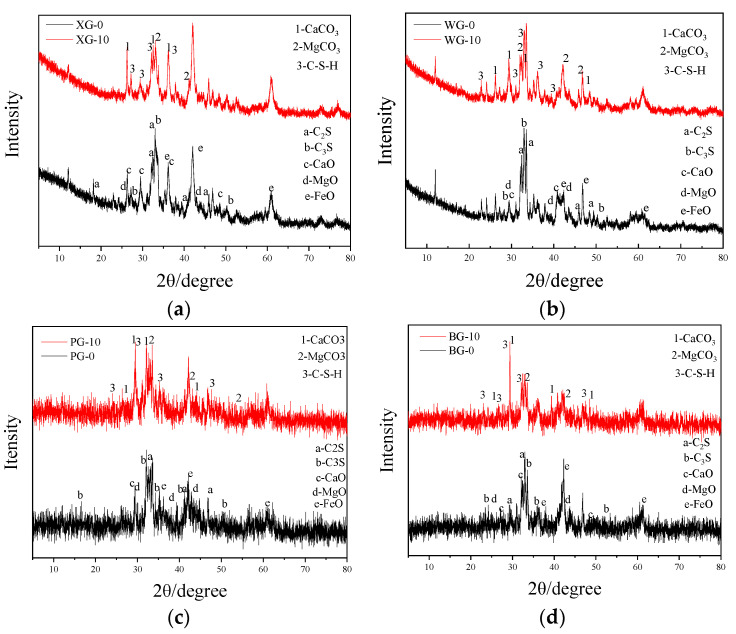
XRD results of steel slags. (**a**) XG steel slag; (**b**) WG steel slag; (**c**) PG steel slag; (**d**) BG steel slag.

**Figure 5 materials-17-01555-f005:**
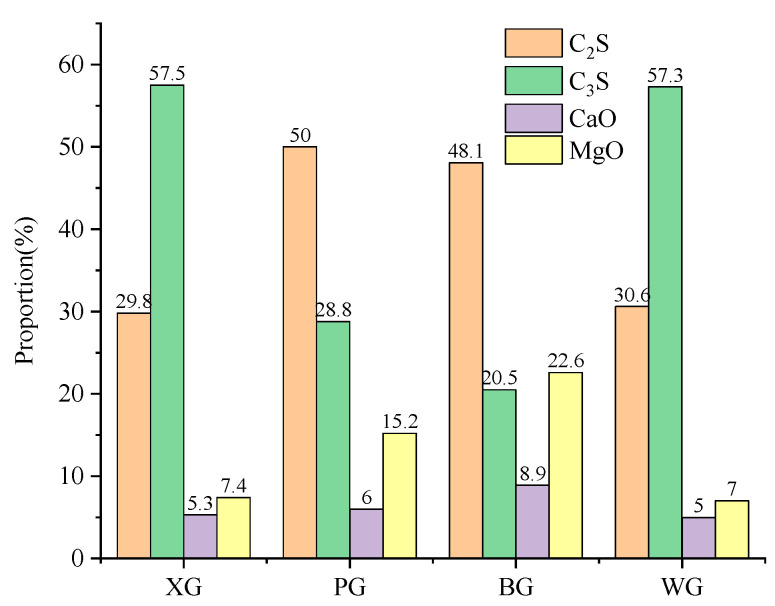
Initial steel slag simple quantitative analysis.

**Figure 6 materials-17-01555-f006:**
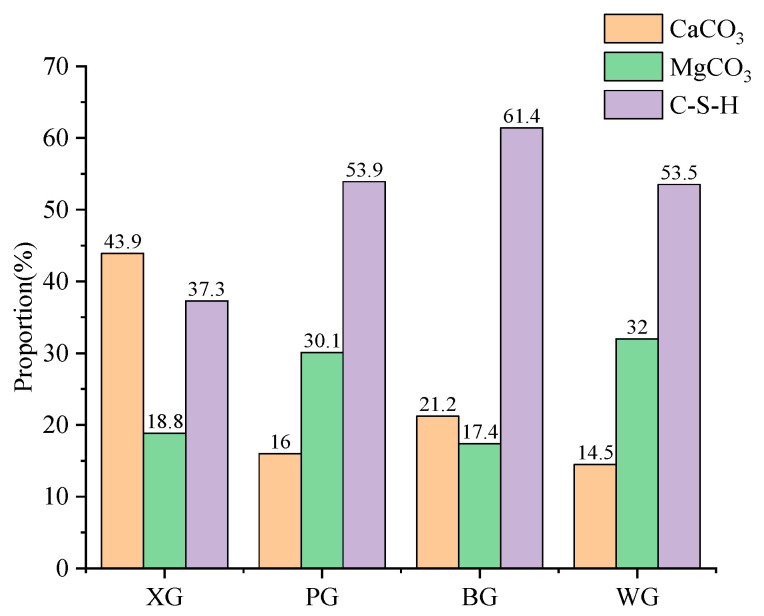
Steel slag simple quantitative analysis after expansion test.

**Figure 7 materials-17-01555-f007:**
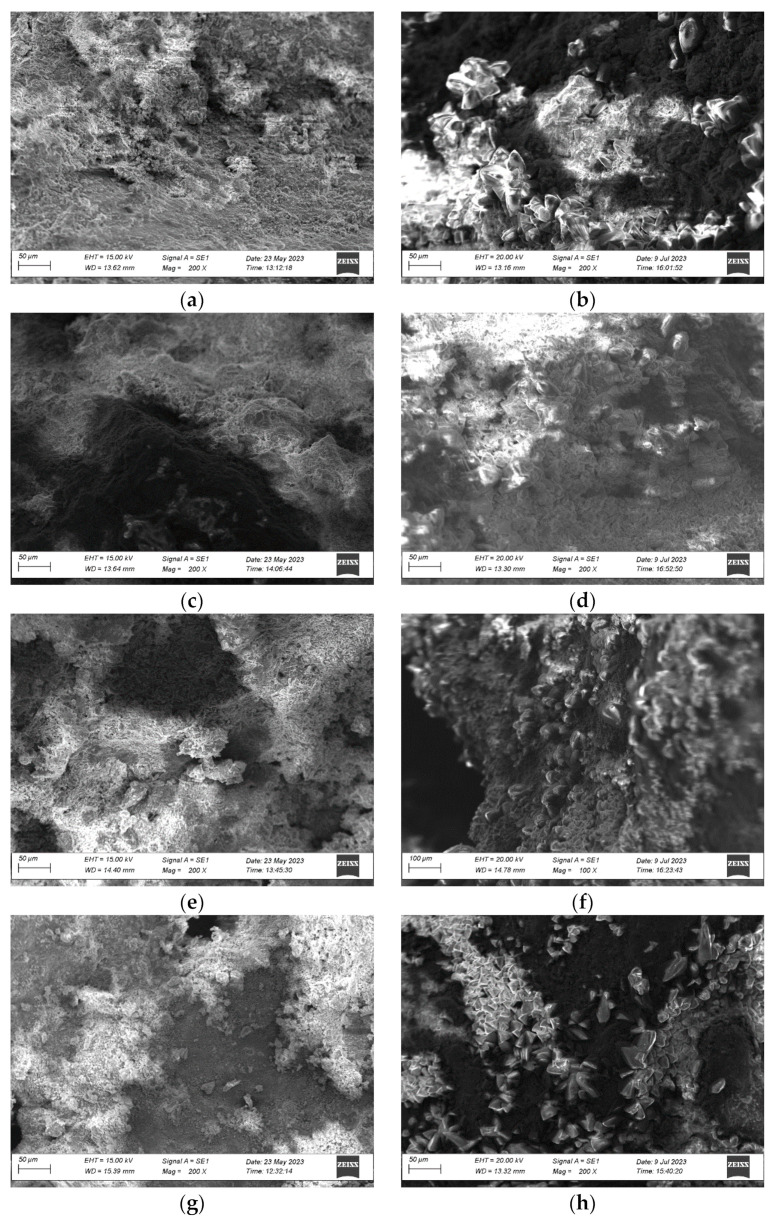
Before and after the expansion test of four kinds of steel slag. (**a**) XG-0; (**b**) XG-10; (**c**) PG-0; (**d**) PG-10; (**e**) BG-0; (**f**) BG-10; (**g**) WG-0; (**h**) WG-10.

**Figure 8 materials-17-01555-f008:**
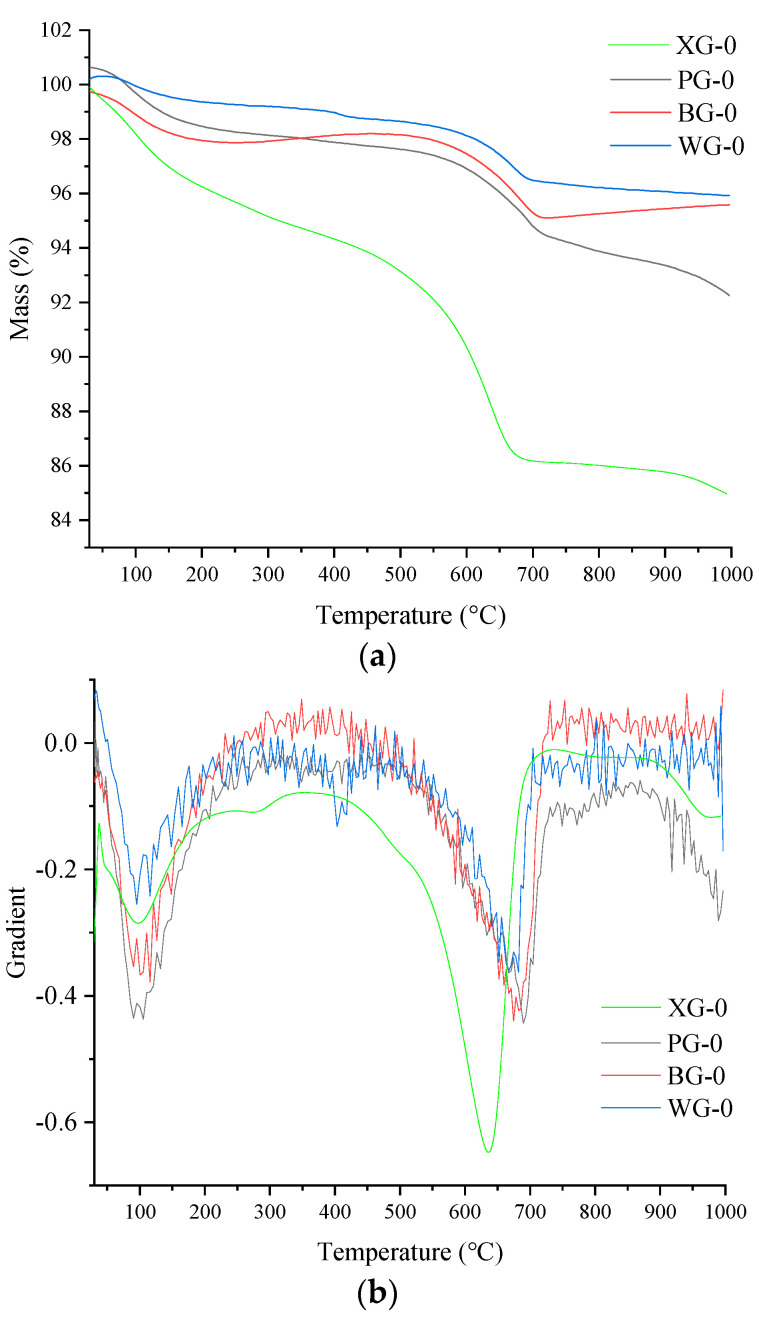
Initial steel slag TG and DTG results. (**a**) TG curve; (**b**) DTG curve.

**Figure 9 materials-17-01555-f009:**
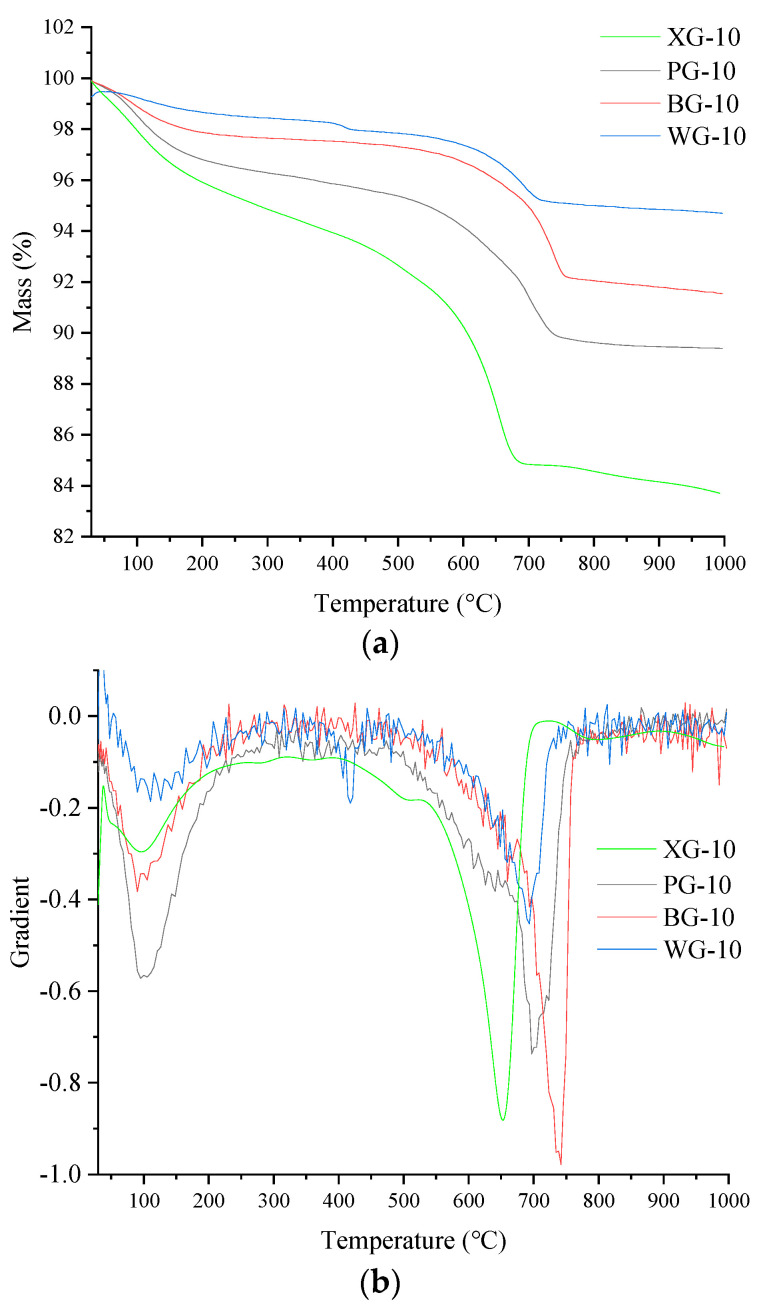
TG and DTG results after expansion test. (**a**) TG curve; (**b**) DTG curve.

**Table 1 materials-17-01555-t001:** Common applications of steel slag [41,42,43,44].

Active Ingredients	Application	Advantages and Disadvantages
C_2_S, C_3_S, C_4_AF, and C_3_A	Replace natural aggregate in concrete	Good durability. Poor water stability
C_3_S and β-C_2_S	Steel slag powder as supplementary cementitious materials	Improve mechanical properties and durability. Potential expansion problem
CaCO_3_, C_2_S and C_3_S	Replace natural aggregate in asphalt mixture	Higher mechanical behavior of mixtures with steel slag aggregate. But require a higher asphalt content

**Table 2 materials-17-01555-t002:** Chemical element composition of the four types of steel slag.

	CaO (%)	Fe_2_O_3_ (%)	SiO_2_ (%)	MgO (%)	MnO (%)	Al_2_O_3_ (%)	P_2_O_5_ (%)	TiO_2_ (%)	Cr_2_O_3_ (%)	SO_3_ (%)
BG	40.80	33.51	13.59	4.39	2.91	1.99	0.99	0.75	0.39	0.23
WG	45.33	31.63	10.46	4.37	3.32	1.51	1.38	0.99	0.43	0.10
PG	46.13	29.85	11.66	4.08	3.65	1.39	1.33	0.93	0.49	0.18
XG	43.36	26.53	13.02	3.92	5.15	0.87	3.68	1.37	0.63	0.74

**Table 3 materials-17-01555-t003:** Density and water absorption.

Types	Parameter	4.75–9.5 mm	9.5–13.2 mm	13.2–16 mm
XG	Density (g/cm^3^)	3.498	3.393	3.377
	Apparent density (g/cm^3^)	3.281	3.192	3.224
	Water absorption (%)	1.89%	1.85%	1.41%
PG	Density (g/cm^3^)	3.488	3.437	3.110
	Apparent density (g/cm^3^)	3.321	3.333	2.865
	Water absorption (%)	1.45%	0.90%	2.75%
BG	Density (g/cm^3^)	3.579	3.628	3.628
	Apparent density (g/cm^3^)	3.351	3.456	3.450
	Water absorption (%)	1.90%	1.38%	1.42%
WG	Density (g/cm^3^)	3.523	3.478	3.512
	Apparent density (g/cm^3^)	3.327	3.343	3.387
	Water absorption (%)	1.67%	1.16%	1.05%

**Table 4 materials-17-01555-t004:** Weigh loss rate of initial steel slag in the specific temperature range.

Temperature (°C)	XG (%)	PG (%)	BG (%)	WG (%)
100–200	1.927	1.228	0.953	0.581
400–500	1.199	0.264	−0.013	0.337
600–750	4.216	2.674	2.314	1.791

**Table 5 materials-17-01555-t005:** The mass loss rate of steel slag in a specific temperature range after the expansion test.

Temperature (°C)	XG (%)	PG (%)	BG (%)	WG (%)
100–200	2.035	1.736	1.015	0.589
400–500	1.289	0.498	0.223	0.398
600–750	5.467	4.336	4.268	2.273

## Data Availability

Data are contained within the article.

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
