# Peer review of "Research on the Properties of Steel Slag with Different Preparation Processes"

_materials, 2024, doi:10.3390/ma17071555_

Round 1

Reviewer 1 Report

Comments and Suggestions for Authors

The study describes the steel slag characteristics after different processing conditions. The quantity of this waste is a growing problem given the increased production. The subject is current. However, there are significant improvements possible.

-          English must be extensively improved. It is hard to read this paper since the sentences are not well structured. Also, the title should be changed. There is no need to mention “material properties” and then steel slag since the slag is a material.

-          Some sentences are too long, and thus the point is lost. For example: “Experts at home and abroad from the physical, chemical and mechanical properties, morphology and other aspects of steel slag to carry out certain experimental work, however, steel slag by the raw materials, steelmaking process, processing technology and other factors and there is an obvious individual variability, the current quantitative characterization of the characteristics of different types of steel slag, different particle size and differential analysis is still not sufficient, the material characteristics are not clear to become a steel slag resource conversion utilization of the lower rate of the The lack of clear material properties is the main reason for the low utilization rate of steel slag resources conversion.”

-          There is no need to use the phrase “solid waste steel slag”.

-          Some sentences are incomplete, and they miss a verb.

-          Often also the meaning is lost. For example, “The average temperature of the steel slag can be cooled to around 50, and then processed”. What kind of processing is meant here?

-          The introduction part, in the concluding paragraph, needs an explanation of the 4 tested slags, not just the abbreviations. What is the process used in those factories, and what are the main differences there? This is explained in more detail in the Materials and Methods section.

-          The experiments section should be rewritten in a way to explain that “this and that” is done and avoid writing a recipe from standards.

-          The tables are missing the units for the parameters shown.

-          It is just water absorption, and not “ratio of water absorption”.

-          In which way was a “ratio” of density presented in Fig. 1 obtained? Why is it calculated to BG? Does this figure have so important practical meaning? Are there any similar results in the literature?

-          Every test done and the results obtained should be compared to the existing literature, if present. Discussions should be widened.

-          In which way was the mixture of different particle sizes of materials determined as suitable for the volume expansion test? Is this test, made for testing soil, also useful for testing steel slag? Are the results comparable to the soil? Find the existing literature.

-          In which way were the peaks obtained by XRD assigned to certain minerals, and are the results comparable to the literature? You can compare your results to mortars in https://doi.org/10.2298/SOS220610014T

-          Explain in detail what is meant under a simple quantitative analysis in expansion test and mineralogy.

-          Chemical analyses are needed to be done for the four slags.

-          I do not believe that you did thermo-gravimetric analysis in Germany, but rather that the instrument is produced there.

-          If mineralogical analysis showed that the slags contained CaCO3, how can you then claim that the dehydroxylation of Ca(OH)2 is the most prominent reaction captured by TGA?

-          The proposition of the possible usage of the 4 tested slags is expected to be written in the conclusion section.

Comments on the Quality of English Language

English is of very low quality and hardly understandable.

Reviewer 2 Report

Comments and Suggestions for Authors

The manuscript shows a characterization of four types of steel slags through density, water absorption, and expansion tests. The physical and mechanical characteristics and volume stability were carried out. This reviewer believes the work must be improved to be published in the Materials Journal.

1) In the introduction, some papers about steel slag and its applications must be cited. I suggest authors read and possibility insert the following papers:
doi.org/10.3390/ma16175841
doi.org/10.1016/j.resconrec.2018.04.023

2) The conclusion needs to answer the question. After all the characterizations are carried out, what applications can these four slags have? Partial replacement of cement, fine or coarse aggregate?
3) I suggest an image of the four types of steel slag in the manuscript.

Reviewer 3 Report

Comments and Suggestions for Authors

This study examines various processed steel slags in China to improve their resource utilization. It analyzes their physical, mechanical, and volume stability properties, as well as mineral phases and thermal stability using various tests. Findings reveal that drum slag has the highest density, a lower expansion rate, and the highest content of free calcium oxide. Different processed steel slags share similar compositions, primarily containing dicalcium silicate, tricalcium silicate, CaO, and MgO. Following expansion tests, the main chemical products include CaCO3, MgCO3, and calcium silicate hydrate. Drum method slag displays the lowest porosity and dense microstructure, with mineral crystals emerging on steel slag surfaces post-expansion test. Thermal stability varies among processing methods. These findings provide insights for effectively selecting and utilizing steel slag resources.

Despite the useful data provided, there are several errors and a lack of evidence demonstrating the originality of the work. Here are some suggestions for enhancing the manuscript:

-The abstract should be improved.

-Spaces between the reference mark and the text should be spaced apart.

-In the introduction, adding a table specifying applications based on the composition of steel slag could provide valuable context.

-Add and justify the originality and importance of this study.

-In the Materials section (2.1), please include X-Ray Fluorescence (XRF) analysis to provide crucial data regarding the initial mineral composition of the steel slag samples. A scheme for producing different steel slag would be useful.

-Why were only four samples used? Is this representative of China's steel slag manufacturing?

-In the Experiments section (2.2), international standards are missing in many cases. Furthermore, equipment marks are deficient.

-The discussion below the results is lacking; there is no comparison with similar results from the literature.

-In the Mineral changes section (3.3), you mentioned that conditions for high-temperature melting are very similar to the process of manufacturing cement. What are these conditions? Can you also add phase diagrams?

-In the Simple quantitative analysis section (3.4), add a reference for the Jade software.

-Add future prospects in the conclusion.

Comments on the Quality of English Language

Moderate editing of English language required

Round 2

Reviewer 1 Report

Comments and Suggestions for Authors

The paper seems sufficiently improved to be published.

Comments on the Quality of English Language

English is improved.

Reviewer 2 Report

Comments and Suggestions for Authors

This reviewer thanks the authors for their effort in revising the manuscript. The authors answered all my questions. The paper has been improved, and this reviewer recommends publishing this content in the Materials Journal.